# Chemical Intolerance and Mast Cell Activation: A Suspicious Synchronicity

**Raymond F. Palmer** [1,*], **Tania T. Dempsey** [2] and **Lawrence B. Afrin** [2]

1 Department of Family and Community Medicine, University of Texas Health Science Center at San Antonio, San Antonio, TX 78229-3900, USA

2 AIM Center for Personalized Medicine, Purchase, NY 10577, USA; drdempsey@aimcenterpm.com (T.T.D.); drafrin@aimcenterpm.com (L.B.A.)

* Correspondence: palmerr@uthscsa.edu; Tel.: +1-210-827-7681

**Abstract:** *Background*: Chemical Intolerance (CI) is characterized by intolerances for chemicals, foods, and drugs with multi-system symptoms. As yet, the biomechanism remains unclear. One study reported converging lines of evidence supporting a substantive association between mast cell activation syndrome (MCAS) and CI. The purpose of this study is to (1) confirm a previous report demonstrating that 60% of MCAS patients report CI and (2) examine the parallels between symptoms and intolerances in CI and MCAS. *Methods*: Five hundred forty-four MCAS patients were assigned a clinical MCAS score using a validated assessment instrument and were assessed for CI using the validated Quick Environmental Exposure Sensitivity Index. *Results*: Our outcomes confirm the previously published study where the majority of MCAS patients also have CI. There was a clear overlap between various ICD-10 diagnostic categories and CI symptoms, providing further support for a potential shared mechanism. *Conclusions*: Exposures to pesticides, volatile organic compounds, combustion products, and mold have previously been reported as initiators of CI. However, until recently, little was known about the biological mechanism involved that could explain the multisystem symptoms associated with CI. This paper addresses a newly identified biomechanism for disease, which may underlie a host of "medically unexplained symptoms" triggered by xenobiotics.

**Keywords:** chemical intolerance; toxicant-induced loss of tolerance; mast cell activation syndrome (MCAS)





## 1. Introduction

*Chemical Intolerance (CI):* CI is characterized by multi-system symptoms and intolerances for chemical inhalants, foods/food additives, and drugs. Prevalence estimates vary according to whether it is clinically diagnosed (0.5–6.5%) or self-reported (average of ~20%) [1–5]. Researchers in the US and Japan have reported increased prevalence rates over a 10-year period [6,7].

In prior work, we have described CI initiation by acute high-level exposure to a toxicant such as pesticide application, chemical release, or repeated chronic lower-level exposures to toxicants such as volatile organic compounds (VOCs) in a "sick" building [8–10]. However, the biomechanism for this condition remains unclear. Notwithstanding, a general disease mechanism called Toxicant-Induced Loss of Tolerance (TILT) explains the initiation, symptoms, and intolerances to chemicals, foods, and medicines reported worldwide by individuals with this condition [8–10]. To be clear, we consider CI to be the condition or the medical disorder and TILT the mechanism that results in CI.

As an explanatory mechanism for CI, TILT is a two-stage process involving an initiating exposure followed by triggering of new-onset intolerances by subsequent exposures to previously tolerated and/or structurally unrelated chemical inhalants, ingestants, and medications [8–10]. As depicted in Figure 1, symptoms of CI commonly include fatigue, headache, weakness, and rashes; involve muscles and joints, the digestive system, and the

respiratory tract; and often affect mood, memory, and concentration [8–11] Large numbers of CI patients attribute their illness to a well-defined exposure event such as exposures to pesticides, new construction or remodeling, indoor air contaminants, or a flood- or water-damaged building resulting in mold and bacterial growth [12–14] (see Figure 2).

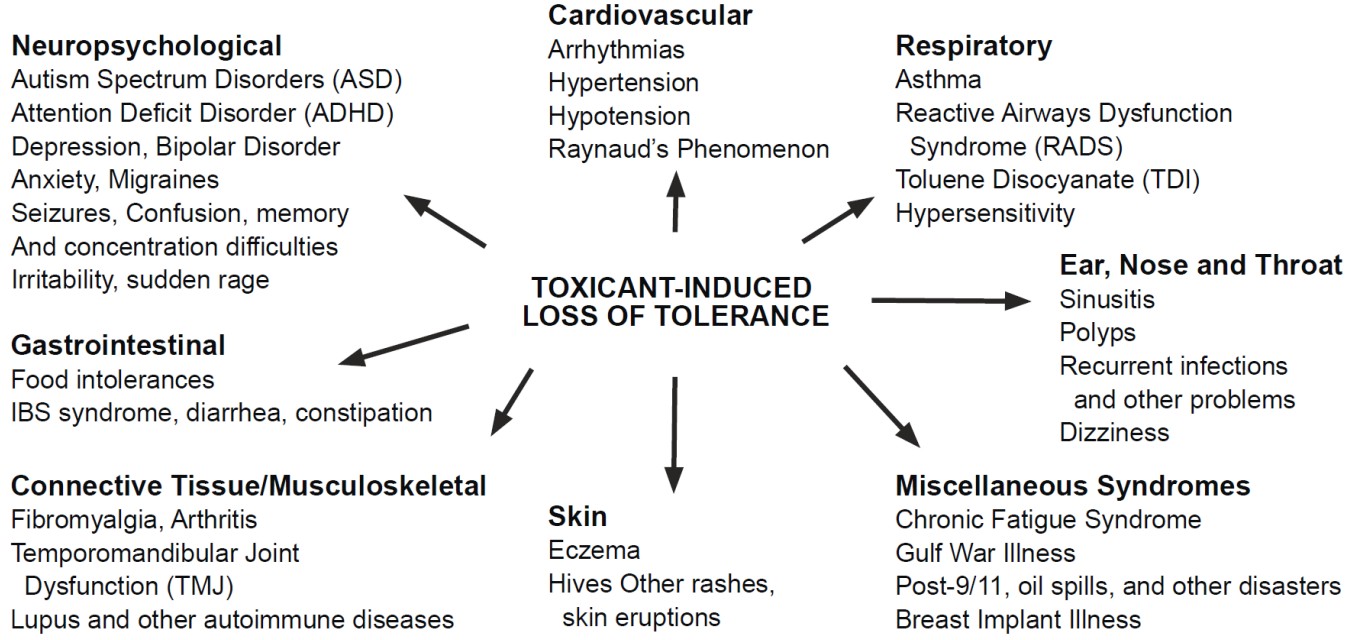

**Figure 1.** Conditions and symptoms of CI/TILT.

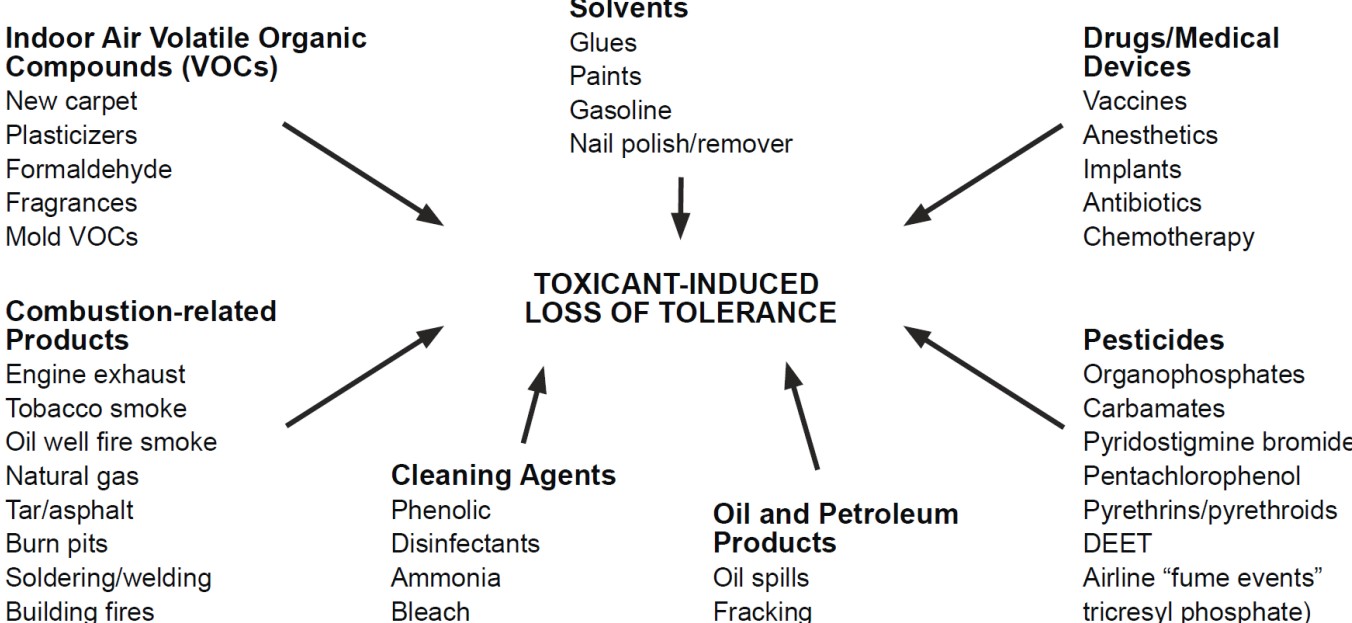

**Figure 2.** Potential initiators and triggers of CI/TILT.

Assessing CI often involves using the Quick Environmental Exposure and Sensitivity Inventory (QEESI) [15], a 50-item validated questionnaire designed to assess intolerances to inhaled chemicals, foods, and/or drugs. To date, researchers in more than a dozen countries have used the QEESI because it offers high sensitivity and specificity for differentiating individuals with CI from the general population [16–18] (see Palmer et al., 2021, for a comprehensive list of studies using the QEESI in 16 countries [3]).

### 1.1. Biological Correlates of CI

Several studies suggest that CI results from the interaction between alterations in the redox system, glutathione depletion, and pro-inflammatory cytokines affecting the expression of metabolizing and antioxidant enzymes [19–21]. Confirmatory studies, however, have been inconsistent [22], with a poor understanding of why.

Targeted genetic association studies have focused on (a) inflammatory and oxidative stress pathways [23]; (b) genes coding for enzymes that metabolize xenobiotics (e.g., SOD, NAT) [24–26]; (c) genetic polymorphisms that involve xenobiotic detoxification processes such as phase I and II enzymes [27,28]; (d) cytochrome P450 isoenzymes involved in metabolizing drugs [29]; and (e) genes such as PON1/PON2 involved in the detoxification of organophosphate pesticides [26,30]. However, reports from other researchers have been inconsistent [23,31,32]. Relevant to research on CI, Fujimori et al. [33] targeted specific genotypes and assessed CI using the QEESI in 1084 employees of Japanese companies. Comparing those with and without CI, no significant differences were found in the allelic distribution of genetic polymorphisms.

Berg et al. [30] assert that the inconsistent findings in the literature may be the result of gene–environment interactions, genetic heterogeneity in CI, small sample sizes, and/or methodological factors such as the lack of standardized laboratory and/or assessment protocols. Conclusions about the physiological elements associated with CI remain uncertain. As such, Vadala et al. [34] propose four levels of CI testing to guide clinicians and to elucidate CI's pathology through a combination of multiple methods, including quantifiable blood tests, improved diagnostic tools, genetic testing, and thorough clinical observation of symptoms.

Miller et al. [35] reported converging lines of evidence supporting a substantive association between mast cell activation syndrome (MCAS) and CI. They proposed mast cell activation as a "plausible and researchable" novel mechanism to explain CI, i.e., if MCAS and CI are closely associated, they would share similar pathophysiologies and exhibit parallel symptoms and intolerances. In that study, it was found that nearly 60% of MCAS patients screened positive for CI using the QEESI. Also, the predicted probability of CI increased as scores on a validated, published protocol used for assessing MCAS increased. Compared to the lowest quartile of MCAS scores, patients in the 2nd quartile were 2.6 times more likely to have CI ($p = 0.027$). Those in the 3rd quartile of MCAS scores were 6.0 times more likely to have CI ($p = 0.0001$), and those in the 4th quartile of MCAS scores were 6.2 times more likely to have CI ($p = 0.0001$) [35].

### 1.2. Mast Cells and MCAS

There is considerable literature describing mast cells (MCs) and their function [36–42]. Briefly, they are sentinel immunity cells that respond to most bodily invasions and insults. MCs originate in the bone marrow and migrate principally to the interface between our tissues and the external environment. They have also been reported to come from progenitor cells made in the extra-embryonic yolk sac [43].

Once triggered via any of their large arrays of receptors and activation pathways [39], MCs release variable subsets from a repertoire of hundreds of distinct mediators [38], resulting primarily in inflammation, allergic-like phenomena, and altered tissue growth and development [39,40]. MCs respond to a wide variety of antigenic triggers, causing the release of mediators particular to the insult and its anatomic location [41,42,44]. The effects of MC mediators can be local, but also, via various mechanisms, can be distant.

Appropriate MC mediator release helps tissues resist and recover from insults. Persistent aberrant release, however, is harmful in ways specific to the locations and patterns of the released mediators. Contemporary exposures (i.e., novel from an evolutionary perspective) may be some of the factors provoking MCs to continually release inflammatory mediators, resulting in a newly recognized condition now termed MCAS [42–49]. Multiple studies consistently suggest that the chronic over-reactive MC mediator production and release in most MCAS patients, both at baseline and in reaction to triggers, are driven principally by a large menagerie of (mostly somatic) variants in regulatory genes within the dysfunctional MCs [45].

Diagnosis of MCAS typically requires (1) chronic and/or recurrent symptoms consistent with aberrant MC mediator release, (2) exclusion of other conditions that might better explain the patient's symptoms, and (3) laboratory evidence of MC activation (i.e., MC mediator release). With a reported prevalence of 10–17%, many MCAS patients respond to treatments targeted at MCs and/or the mediators they release [50,51].

This follow-up study, based on the previous study by Miller et al. [35], adds a new cohort of MCAS patients in order to (1) confirm the previous findings demonstrating that 59% of MCAS patients have CI, and combining cohorts, (2) use ICD-10 diagnosis codes to examine the previously observed parallels between symptoms and intolerances in both CI and MCAS.

## 2. Materials and Methods

For this paper, the MCAS group consists of two cohorts for a total of 544 distinct patients seen by authors L.B.A. and T.T.D. Data from the first cohort (n = 149) were collected between September 2017 and April 2018 [35]. Patients seen between May 2018 and September 2021 comprise the second cohort (n = 395). All patients were assigned a clinical score reflecting their likelihood of having MCAS using a validated published MCAS assessment instrument [46,52,53] (e.g., obtaining medical history and physical exam, exclusion of diagnoses better accounting for their many issues, and any laboratory evidence of MC activation). ICD-10 diagnosis codes for these patients (as assigned during routine clinical care by authors L.B.A. and T.T.D.) were also retrieved.

*Quick Environmental Exposure Survey Instrument*

Patients also completed the QEESI [16–18], which has four scales: Symptom Severity, Chemical Intolerances, Other Intolerances, and Life Impact. Each scale item is scored from 0 to 10 (0 = "not a problem" to 10 = "severe or disabling problem"). Total scale scores range from 0 to 100. There is also a 10-item Masking Index, which gauges ongoing exposures (such as to caffeine, tobacco, drugs, fragrances, or pesticides) that can reduce or mask individuals' awareness of their intolerances [54]. The Life Impact scale assesses the degree to which CI affects quality of life, and the Other Intolerances scale includes potential intolerances a person may react to that is not on the Chemical scale. For more detailed explanations of these scales, see Miller and Prihoda [15,16]. Consistent with the large, published literature that uses the QEESI to assess CI [3], only the Chemical and Symptom scales are used. The other scales are shown in this study for cohort comparison purposes only and are discussed in more detail in a previous paper comparing MCAS and other groups [35].

Items on the Chemical Exposure scale ask participants to rate (on a scale from 1 to 10) whether specific chemical exposures (e.g., gasoline, tar, perfume, new furniture, tobacco smoke, paint, cleaning products, and nail polish) would make them feel sick, for example, "you would get a headache, have difficulty thinking, feel weak, have trouble breathing, get an upset stomach, feel dizzy, or something like that".

The Symptom Severity scale asks about common symptoms the person may have, not necessarily associated with the specific exposures on the Chemical Intolerance scale. Table 1 shows the 10 symptoms evaluated on the Symptom scale [15,16].

**Table 1.** Symptoms evaluated on the QEESI symptom scale.

| | |
|---|---|
| 1. | Musculoskeletal Symptoms (MS) Problems with your muscles or joints, such as pain, aching, cramping, stiffness, or weakness. |
| 2. | Airway or Mucous Membrane Symptoms (AIR/MM) Problems with burning or irritation of your eyes, or problems with your airway or breathing, such as feeling short of breath; coughing; or having a lot of mucus, postnasal drainage, or respiratory infections. |
| 3. | Heart/Chest-related Symptoms (COR) Problems with your heart or chest, such as a fast or irregular heart rate, skipped beats, your heart pounding, or chest discomfort. |
| 4. | Gastrointestinal Symptoms (GI) Problems with your stomach or digestive tract, such as abdominal pain or cramping, abdominal swelling or bloating, nausea, diarrhea, or constipation. |
| 5. | Cognitive Symptoms (COG) Problems with your ability to think, such as difficulty concentrating or remembering things, feeling spacey, or having trouble making decisions. |
| 6. | Affective Symptoms (AFF) Problems with your mood, such as feeling tense or nervous, irritable, depressed, having spells of crying or rage, or loss of motivation to do things that used to interest you. |
| 7. | Neuromuscular Symptoms (NM) Problems with balance or coordination, with numbness or tingling in your extremities, or with focusing your eyes. |
| 8. | Head-related Symptoms (HEAD) Problems with your head, such as headaches or a feeling of pressure or fullness in your face or head. |
| 9. | Skin-related Symptoms (SKIN) Problems with your skin, such as a rash, hives, or dry skin. |
| 10. | Genitourinary Symptoms (GU) Problems with your urinary tract or genitals, such as pelvic pain or frequent or urgent urination (for women: or discomfort or problems with your menstrual period. |

Scores of 40 or more on both the Chemical and Symptom scales are considered "*Very suggestive*" of CI. A score of 40 and above on either the Chemical or the Symptom scale alone is considered "*Suggestive*" of CI. Scores below 20 on both scales are considered "*Not suggestive*" of CI—creating 3 potential groups.

Clinical scores for the MCAS patient group were used to predict CI status using a logistic regression model. Analyses were performed using SAS software V9.4 [55]. This study was approved by the University of Texas Health Science Center's San Antonio Institutional Review Board (approval number HSC2021062HR).

## 3. Results

Table 2 shows the two cohorts of MCAS patients, where there is a predominance of females (in line with multiple published estimates consistently showing a roughly 4:1 female: male ratio in the Western first-world populations surveyed thus far [39]). There were no statistical differences in age or gender between the two cohorts (ANOVA, *p* = 0.41; *p* = 0.34, respectively).

In the first cohort of MCAS patients, as reported by Miller et al. [35], 60% of the 149 patients were classified as *very suggestive* of CI, and 37% were classified as *suggestive* of CI according to QEESI criteria [35]. In the second cohort of MCAS patients, 50% were classified as *very suggestive* and 46% as *suggestive*. There was no statistical difference in QEESI scale scores between cohorts (*p* form <0.60.

Table 2 also shows that the average QEESI scale scores differ by the QEESI group categories as defined above. Since the Chemical Intolerance and Symptom scales define these categories as described above, this is obvious. Interestingly, the Life Impact and Other Intolerance scale scores also differ by group, indicating a greater overall illness burden in the highest QEESI categories relative to the lower categories. Further, average MCAS scores also increase in the same manner, i.e., they are higher as QEESI-defined CI severity increases.

**Table 2.** QEESI scores categorizing three distinct chemical intolerance groups among MCAS patients.

| | Total Sample (N = 544) | | | | | | | |
|---|---|---|---|---|---|---|---|---|
| | Cohort 1 (N = 149) | | | | Cohort 2 (N = 395) | | | |
| | Mean | Std Dev | Min | Max | Mean | Std Dev | Min | Max |
| Age | 46.04 [ns] | (13.76) | 21.00 | 80.00 | 44.48 | (14.85) | 18.00 | 83.00 |
| % Female | 86.58 [ns] | - | - | - | 82.03 | - | - | - |
| QEESI CI Categories and Scale Scores | | | | | | | | |
| Not Suggestive | (N = 4) 3% | | | | (N = 13) 3% | | | |
| Chemical | 2.3 [ns] | (2.6) | 0 | 6 | 6.1 | (7.3) | 0 | 18 |
| Symptoms | 15.5 [ns] | (2.1) | 13 | 18 | 15.7 | (3.4) | 8 | 20 |
| Other Intolerances | 13.0 [ns] | (8.1) | 3 | 21 | 8.9 | (8.9) | 0 | 28 |
| Life Impact | 26.5 [ns] | (10.7) | 12 | 37 | 12.3 | (13.1) | 0 | 36 |
| Masking | 2.3 [ns] | (0.6) | 2 | 3 | 3.8 | (2.3) | 1 | 9 |
| MCAS Score | 13.8 [ns] | (7.9) | 3 | 21 | 9.3 | (4.5) | 2 | 17 |
| Suggestive | (N = 55) 37% | | | | (N = 183) 46% | | | |
| Chemical | 22.1 [ns] | (17.6) | 0 | 66 | 21.8 | (18.9) | 0 | 92 |
| Symptoms | 44.2 [ns] | (17.9) | 10 | 87 | 46.8 | (17.3) | 12 | 100 |
| Other Intolerances | 33.2 [ns] | (17.8) | 0 | 80 | 33.1 | (17.0) | 0 | 88 |
| Life Impact | 40.8 [ns] | (27.3) | 0 | 100 | 38.5 | (25.3) | 0 | 100 |
| Masking | 3.3 [ns] | (1.8) | 0 | 8 | 2.9 | (1.5) | 0 | 7 |
| MCAS Score | 18.1 [ns] | (9.5) | 2 | 44 | 15.9 | (7.9) | 0 | 46 |
| Very Suggestive | (N = 90) 60% | | | | (N = 199) 50% | | | |
| Chemical | 65.6 [ns] | (16.8) | 40 | 100 | 67.9 | (17.1) | 40 | 100 |
| Other Intolerances | 63.5 [ns] | (12.7) | 40 | 89 | 69.4 | (15.2) | 40 | 100 |
| Symptoms | 56.6 [ns] | (15.8) | 10 | 96 | 56.3 | (17.3) | 13 | 100 |
| Masking | 70.7 [ns] | (21.5) | 26 | 100 | 71.4 | (22.1) | 4 | 100 |
| Life Impact | 2.5 [ns] | (1.4) | 0 | 7 | 2.7 | (1.6) | 0 | 7 |
| MCAS Score | 23.5 [ns] | (8.7) | 3 | 49 | 24.8 | (9.7) | 1 | 58 |

[ns] No significant age or differences between groups (ANOVA, $p = 0.41$; $p = 0.34$, respectively). No significant difference between groups on all QEESI scales and MCAS scores ($p \leq 0.50$ for all scales). As expected by design, there are significant differences between QEESI categories for both categories (ANOVA, $p < 0.0001$). Masking index is significantly lower in the very suggestive group (ANOVA, $p = 0.0009$).

The Pearson correlation coefficient between MCAS scores and the Chemical Intolerance scale score is 0.42 and 0.51 for the Symptom scale score. In a logistic model, MCAS scores were used to predict CI status adjusted for age and gender. Because there were only 17 total cases *Not Suggestive* of CI in this sample with valid MCAS scores, we compared MCAS scores between the *Very Suggestive* and *Suggestive* categories only. For each one-unit increase in MCAS score, there is an 11% increase in the odds of being categorized as *Very Suggestive* vs. *Suggestive* (OR = 1.11, 95% Confidence Interval = 1.08–1.13, $p < 0.0001$). Figure 3 displays the increasing probability of CI given increases in MCAS scores.

The predicted probability of CI as a function of MCAS scores is derived from the logistic regression model. The small circles along the top and bottom of the graph are the observed data points (*Very Suggestive* of CI at the top and *Suggestive* along the bottom). Multiple patients may cluster at any given MCAS score. The solid line is the prediction curve showing that the probability of CI (*Y* axis) increases rapidly as MCAS scores increase. For each one-unit increase in MCAS score, there is an 11% increase in the odds of being *Very Suggestive* vs. *Suggestive* (OR = 1.11, 95% Confidence Interval = 1.08–1.13, $p < 0.0001$).

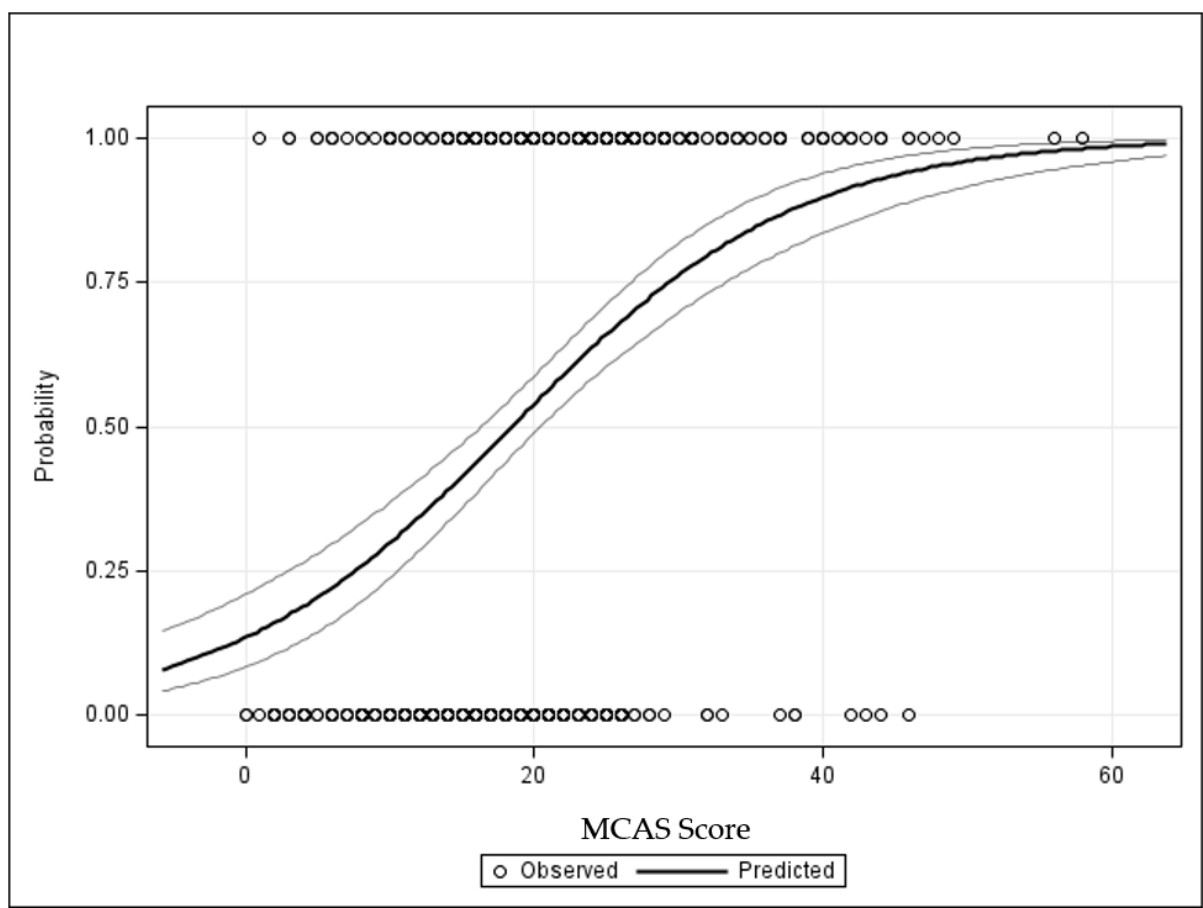

**Figure 3.** Probability of chemical intolerance given MCAS score.

Table 3 shows that among the 544 MCAS patients, there were 2484 ICD-10 diagnoses that were coded in the patient charts, with an average of 4.6 diagnoses per patient. This snapshot of the number of diagnoses is likely significantly lower than the true average number of diagnoses given to these patients over their lifetimes, given the multisystem nature of MCAS. Generally, there is a long path before most MCAS patients receive an underlying diagnosis of MCAS. Along that path, they accumulate many "superficial", non-etiologic diagnoses [56]. Furthermore, due to time constraints, the physicians who provided the data for this study (L.B.A. and T.T.D.) tended to code only what they saw as the patients' root diagnoses (such as MCAS and chronic active infections) as opposed to coding all the (mostly MCAS-driven) diagnoses each patient had received.

**Table 3.** ICD-10 diagnoses received by MCAS patients.

|  | N | (%) |
|---|---|---|
| INFECTION (59%) |  |  |
| Lyme disease, unspecified | 80 | 12% |
| Bartonella | 72 | 11% |
| Other protozoal diseases, not elsewhere classified | 63 | 10% |
| Viral infection, unspecified | 54 | 8% |
| Systemic Bartonellosis | 38 | 6% |
| Bacterial intestinal infection, unspecified | 29 | 5% |
| Candidiasis, unspecified | 28 | 4% |

**Table 3.** *Cont.*

|  | N | (%) |
|---|---|---|
| Bacterial infection, unspecified | 11 | 2% |
| Unspecified infectious disease | 7 | 1% |
| MUSCULOSKELETAL (52%) | | |
| Chronic fatigue, unspecified | 220 | 34% |
| Other fatigue | 103 | 16% |
| Muscle weakness (generalized) | 10 | 2% |
| ENVIRONMENT (47%) | | |
| Other adverse food reactions, not elsewhere classified | 202 | 31% |
| Contact with and (suspected) exposure to mold (toxic) | 60 | 9% |
| Adverse effect of other drugs, medicaments, and biological substances, sequela | 47 | 75% |
| NUTRITIONAL DEFICIENCIES (41%) | | |
| Vitamin D deficiency, unspecified | 60 | 9% |
| Vitamin B12 deficiency anemia, unspecified | 50 | 8% |
| Iron deficiency anemia, unspecified | 52 | 8% |
| Vitamin deficiency, unspecified | 48 | 7% |
| Essential fatty acid (EFA) deficiency | 23 | 4% |
| Folic acid deficiency anemia NOS | 15 | 2% |
| Other vitamin B12 deficiency anemias | 15 | 2% |
| GASTROINTESTINAL (41%) | | |
| Diarrhea, unspecified | 55 | 9% |
| Generalized abdominal pain | 51 | 8% |
| Nausea | 47 | 7% |
| Unspecified abdominal pain | 25 | 4% |
| Nausea and vomiting | 21 | 3% |
| Abdominal distension (gaseous) | 16 | 2% |
| Gastro-esophageal reflux disease without esophagitis | 16 | 2% |
| Other irritable bowel syndrome | 16 | 2% |
| Gastroparesis | 11 | 2% |
| Chronic idiopathic constipation | 8 | 1% |
| SKIN (33%) | | |
| Angioneurotic edema, initial encounter | 44 | 7% |
| Rash and other nonspecific skin eruption | 47 | 7% |
| Flushing | 43 | 7% |
| Idiopathic urticaria (Hives) | 38 | 6% |
| Other pruritus | 40 | 6% |
| CARDIOVASCULAR (24%) | | |
| Orthostatic hypotension | 51 | 8% |
| Palpitations | 34 | 5% |
| Metabolic syndrome | 33 | 5% |
| Dizziness and giddiness | 35 | 5% |

**Table 3.** *Cont.*

|  | N | (%) |
|---|---|---|
| COGNITIVE (23%) |  |  |
| Mild cognitive impairment, so stated | 146 | 23% |
| NEUROMUSCULAR (15%) |  |  |
| Chronic pain syndrome | 98 | 15% |
| AFFECT (15%) |  |  |
| Insomnia | 30 | 5% |
| Anxiety disorder, unspecified | 34 | 5% |
| Generalized anxiety disorder | 21 | 3% |
| Major depressive disorder, recurrent, unspecified | 10 | 2% |
| HEAD (13%) |  |  |
| Migraine | 23 | 4% |
| Other headache syndrome | 59 | 9% |
| GENETIC (8%) |  |  |
| Methylenetetrahydrofolate reductase deficiency | 53 | 8% |
| ENDOCRINE (7%) |  |  |
| Hypothyroidism, unspecified | 46 | 7% |
| AIR/MUCOUS MEMBRANE (7%) |  |  |
| Dyspnea, unspecified | 46 | 7% |
| Total | 2484 |  |

Note: Additive percentages are over 100% due to the great majority of patients receiving more than one diagnosis.

The ICD-10 coded diagnoses have been grouped by the type of diagnosis in Table 3. Note that 9 of these associated 14 classifications match the categories of the QEESI Symptom scale (Table 1): Musculoskeletal, Gastrointestinal, Skin, Cardiovascular, Cognitive, Affective, Neuromuscular, Head, and Airway/Mucous Membrane. Figure 4 graphically depicts the distribution of these diagnoses in the combined cohorts and provides another visual demonstrating the substantial overlap of symptoms with the QESSIs symptoms from Table 1.

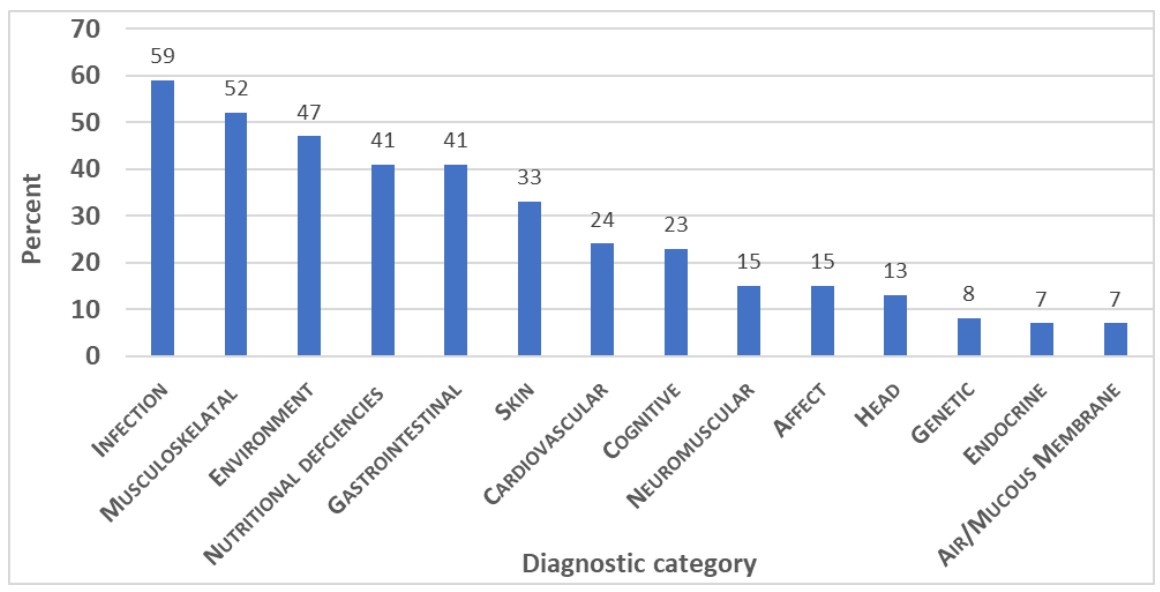

**Figure 4.** System categories of ICD-10 diagnosis among MCAS patients.

## 4. Discussion

Exposures to pesticides, volatile organic compounds (VOCs) during new construction/remodeling, combustion products, mold, and fragranced personal care and household products have previously been reported as initiators of CI [8,11,12,17,18]. However, until recently, little was known about the biological mechanism(s) that could explain the development of these patients' multisystem symptoms resulting from such exposures associated with CI [12,17,18,57]. In this study, we confirm the observed association between MCAS and CI first proposed by Miller et al. [35]. The findings from this study, using a much larger cohort, are consistent with the previously published study of the initial cohort of 149 MCAS patients.

However, because both samples were from the same clinic, our understanding of the strong association we observed between MCAS patients should be independently replicated by other clinicians and appropriate animal and benchtop studies.

Comparing Tables 1 and 3, there is a clear overlap between the various ICD-10 diagnostic categories in these MCAS patients and QEESI symptom categories (e.g., Head-related, GI, Cognitive, Cardiovascular, Skin, and Neuromuscular), providing further support for a shared mechanism. The rates of patients with Infection in Table 3 (59%) are high enough to question whether a large number of the patients studied in this paper might have secondary MCAS (i.e., MCAS solely reactive to infection rather than primary MCAS complicated/aggravated by infection). However, the fact that most patients appear to have suffered for decades with multisystem inflammatory, allergic, and dystrophic issues consistent with the behavior of MCAS (i.e., dating back well before any infection likely would have occurred) makes clear these infections were not life-long primary issues driving purely secondary MCAS but rather were aggravants acquired at various points in the patients' courses with their primary MCAS.

The practice used as the source of the data in this study is located in New England, where tick-borne infections are common. The established literature shows that some of the most common tick-borne infectants clearly have the capacity for driving mast cell activation [58]. It is not surprising that some non-trivial proportion of the practice's MCAS population has also been found to bear proven cases of tick-borne infections. Furthermore, as the literature is also clear that cases of coincident tick-borne infections (i.e., simultaneous infection with more than one tick-borne microorganism) are not uncommon, it is not surprising that a number of cases in the dataset indeed are cases of coincident infection. In fact, there is substantial overlap in the subpopulation bearing proven Borrelia infection with the subpopulation bearing proven Bartonella infection.

We should be careful to note that the full histories in the infected patients strongly suggest that MCAS predated any infections in them, i.e., their MCAS likely is their primary issue, and their infections, which developed later, have been serving as additional "triggers" in them no more or less significant than any other triggers, including assorted provocative frequent or chronic chemical exposures.

After trigger identification and avoidance strategies are implemented, potential medical interventions for CI may include many of those used to treat MCAS, such as H1 and H2 histamine receptor antagonists and cromolyn [51]. Patients whose dysfunctional mast cells are triggered by excipients in commercially available formulations of relevant drugs may require compounded formulations [52]. The literature has shown that even entire papers focused exclusively on the treatment have found it difficult to capture the full range of such treatment [50].

The historic lack of understanding of the underlying biomechanisms for CI has inappropriately fostered a view of CI as a psychosomatic disorder (e.g., Multiple Chemical Sensitivity) [59]. However, the psychosomatic view of CI has diminished over time due in part to a greater understanding of the roles of gene–environment interactions, oxidative stress processes, olfactory and sensory pathways, and systemic inflammation in CI [60–63]. In addition to avoiding symptom triggers, other treatment options have been identified, and while evidence-based treatments are not yet available, multidisciplinary integrative

care models have been suggested [64–68]. Fares-Medina et al. [69] suggest that identifying patterns of symptoms by age group and gender will allow for earlier diagnoses and improve prognosis and treatment.

*Mast Cells as a Potential Biomechanism in CI*

In earlier papers, it was proposed that toxicants appear to alter MCs, possibly epigenetically [45], resulting in MCs that may react aberrantly to low levels of previously tolerated xenobiotics [1–3]. A broad array of toxicants can potentially initiate CI and are often divided into two broad classes: biogenic toxicants and anthropogenic toxicants. Biogenic initiators include particles or VOCs arising from toxic molds or algae. Frequently cited anthropogenic initiators are particles or gases derived from fossil fuels—that is, coal, oil, or natural gas, their combustion products, and their synthetic chemical derivatives. The latter include pesticides, plasticizers, endocrine disrupters, dioxins, and all so-called persistent organic pollutants (POPs) [70]. The greatest exposures to these chemicals occur indoors, where concentrations of volatile synthetic organic chemicals commonly exceed outdoor levels by up to 100 times [71]. The sources include solvents and fragrances released by cosmetics, personal care products, and cleaning and laundry products.

All of these synthetic chemicals are foreign to our evolutionarily ancient MCs. These chemicals enter via every conceivable route—the olfactory–limbic tract, the airways, the gastrointestinal tract, the urogenital tract, the skin, and even via injection/implantation. All of these pathways, as well as our blood and lymphatics, are lined with MCs, which lie in wait until a xenobiotic appears [38,42]. If the MCs in that specific tissue have previously been insulted, whether acutely by a major exposure (e.g., a pesticide application) or perhaps repeatedly at lower exposure levels (e.g., by VOCs in a sick or moldy building), they spring into action upon re-exposure, releasing cascades containing potentially hundreds of mediators designed to protect against the invaders. Inside each MC are granules containing pre-formed mediators. At the same time, the MC begins to manufacture and release specific inflammatory mediators based on prior encounters with the substance [48].

The evolutionary role of MCs is to protect our internal milieu from the external chemical environment, that is, protect us from all xenobiotics or "non-self". MCs are the first responders to any insult. If the insult involves a relatively large molecule or antigen (pollen, animal dander, or vaccine), the MC can initiate humoral or "extrinsic" immunity, leading to the production of immunoglobulins [72]. If the insult involves a small foreign molecule, such as smoke particles resulting from burning a fossil fuel (burn pits or fracking), MCs may initiate cell-mediated or intrinsic immunity, also known as Type IV delayed-type hypersensitivity (DTHS), which may require 48–72 h to manifest [72]. This makes clear that mast cell reactivity to various triggers can contribute to the development of humoral and cellular immunity of all types, which certainly is not surprising given the vast menagerie of mediators available from mast cells.

## 5. Conclusions

This paper confirms the strong likelihood—given the known biological behaviors of mast cells, the known clinical behaviors of MCAS, and prevalent findings of clinical issues of CI in a sizable cohort of MCAS patients—that MCAS may be a key biomechanism for a disease which underlies a host of "medically unexplained symptoms" and syndromes triggered by xenobiotics. MCAS doctors explain these adverse reactions as being due to altered MCs sensitizing and degranulating when provoked by previously tolerated chemicals or physical stimuli [8,12]. Those with CI and/or MCAS present as extraordinarily challenging and complex patients [73]. The revelation of this biomechanism has profound implications for patients and their families, health care providers, public health practitioners, and policymakers.

From a public health standpoint, improved regulation of environmental initiators such as pesticides and combustion products and triggers such as fragranced consumer products and food additives may help reduce the impact of CI and MCAS. Mental health

practitioners (psychiatrists, psychologists, and social workers) need to understand and be able to use the QEESI to assess CI, identify individuals whose problems may stem from toxic exposures, and refer them appropriately. Environmental health professionals (e.g., industrial hygienists and indoor air specialists) are already using the QEESI to guide individuals and families with illnesses related to their exposures. To meet the rapidly evolving needs of 21st-century populations, medical and public health training worldwide needs to incorporate chemical exposures, CI, and MCs in their curricula.

**Author Contributions:** Conceptualization, L.B.A., R.F.P. and T.T.D.; methodology, R.F.P. and L.B.A.; formal analysis, R.F.P.; resources, L.B.A. and T.T.D.; data curation, L.B.A. and R.F.P.; writing—original draft preparation, R.F.P.; writing—review and editing, L.B.A. and T.T.D.; supervision, L.B.A. and R.F.P.; project administration, R.F.P. and L.B.A.; funding acquisition, R.F.P. All authors have read and agreed to the published version of the manuscript.

**Funding:** This research was funded by the Marilyn Brachman Hoffman Foundation, Fort Worth, Texas.

**Institutional Review Board Statement:** This study was conducted in accordance with the Declaration of Helsinki and approved by the Institutional Review Board at the University of Texas Health Science Center at San Antonio, Institutional Review Board (approval number HSC20150821H).

**Informed Consent Statement:** Patient consent was not applicable since non-identifiable anonymous secondary data analysis was used.

**Data Availability Statement:** The data are available upon reasonable request from the first author.

**Conflicts of Interest:** The authors declare no conflict of interest.

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
