# Peer review of "Chemical Intolerance and Mast Cell Activation: A Suspicious Synchronicity"

_jox, doi:10.3390/jox13040045_

Round 1

Reviewer 1 Report

Comments and Suggestions for Authors

This paper describes an association between reports of chemical intolerance and mast cell activation. The aim is to confirm earlier results and to examine the parallels between these two afflictions. The conclusions are that a relationship can be found, and the authors suggest that there is a common mechanism.

Introduction

Row 43-55 and also Figure 1 and Figure 2: Please explain the connection to chemical intolerance, is CI a result of TILT or is TILT another name for CI? Also, I do not understand the figures, figure 1 is showing different comorbidities to TILT, where do CI fit into this picture? Figure 2 presents intiators and triggers for TILT which I do not think are of relevance for the scope of this paper.

Methods

Row 136-140: To be able to assess this part more information is needed about the score, how was it calculated, what is included etc?

Row 142-157: please clarify the description of the scales that are used. I understand that QEESI was used, but you only provide descriptions of 2 of the four subscales (and please be more consistent when naming them, that would make it easier to read).

Very little information is provided about the masking index – what scale, how was the index calculated etc?

Row 153-157: where does the scores for “not suggestive”, “suggestive” and “very suggestive” come from?  Were only results from two of the scales included in this, why?

Table 1. A more thorough description of the questionnaire together with references would be enough and I would prefer to have a table with background information about the cohort.

Results

Row 164-166: please provide more information about the two different cohorts (in the methods section. I suggest you make a table with all information together with statistics. As it is now there are no numbers to backup the p value provided (p<0.07). Please also explain what the QEESI criteria were and how these were used in this study.

Table 2: I suggest that the authors remake the table to make it easier to read, what I mean is that you should have the three groups as columns and the scores on QEESI and the other instruments as rows. Please also provide statistics in the table. As the table is now it is very difficult to compare the groups.

Row 203-217: Was this a part of the aim? I think that this is maybe background information and should not be presented here.

Table 3 is very uninformative, why not divide it into the groups of CI? I do not understand the relevance of this table as it is now.

Figure 4. What is the connection to CI?

Discussion

Row 235- 246: I think that the conclusion about shared mechanism is difficult to draw based on the information provided here. The symptoms in table 1 is very general including symptoms from the whole body and it is therefore not strange that it overlaps with the diagnoses in table 3.

Row 257-300: although an interesting discussion, most of it are not relevant in the context of the aims provided.

Author Response

Word file included

Reviewer 2 Report

Comments and Suggestions for Authors

Palmer, Dempsey, and Afrin present an observational study on MCAS patients using the QEESI to look for alignment between MCAS and CI/TILT diagnoses. This is a well-written and easy to follow paper. It is a powerful contribution because of its contribution of human-based results with a good sample size (n=527 from two combined patient cohorts). The introduction sets an excellent background. Some editors and reviewers might think it is a bit long and overly referenced, but I would disagree and encourage the authors to preserve this essential synthesis of the literature – in my experience knowledge of MCAS is non-existent among most immunologists who are not directly studying it (and there are still physicians who do not recognize it as a legitimate condition); furthermore the background on CI is necessary for the hypothetical linking.

Comments to address:

1.       There seems to be a preponderance of Lyme disease (12%) and Bordetellosis (11%) among the cohort. I understand that these are part of a category regarding infection (59%); however, it was not clear to me whether any of these represented the same patient. Was there any overlap (potential lines of synergy or additive effect) with these patients? Given that these are relatively specific immune interactions.

2.       To the point above, I was surprised there was not an enriched discussion of these and related vectors. I think this should be further synthesized. Such discussion will further support the speculation of a shared CI/MCAS mechanism.

3.       Minor detail: lines 273 -278 makes some important factual statements without citation.

4.       Some potentially provocative statements are made regarding the function of MC. For example, lines 292-294 suggest that mast cells are initiators of IgE-mediated allergy. This would go against the commonly accepted thread of a “silent phase”.

5.       The speculative forward thinking of the discussion and conclusion could also be beefed up with thoughts or potential literature lines on mast cell stabilizers in the context of CI/TILT. Would it be reasonable to assess the use of cromolyn, etc. in these populations if there is a shared mechanism?

6.       Minor typo, missing “II” after “World War” on line 269.

Author Response

word file included.
